# Do we still need a canary in the coal mine for laboratory animal facilities? A systematic review of environmental health monitoring versus soiled bedding sentinels

Megan R. LaFollette[1]*, Caroline S. Clement[2,3], Kerith R. Luchins[4], Christopher A. Manuel[5], Patricia L. Foley[6], Wai H. Hanson[7], Christina Pettan-Brewer[8], Caroline B. Winn[9], Joseph P. Garner[2,10]

1 The 3Rs Collaborative, Denver, Colorado, United States of America, 2 Department of Comparative Medicine, Stanford University, Stanford, CA, United States of America, 3 Department of Biology, Stanford University, Stanford, CA, United States of America, 4 Animal Resources Center and Department of Surgery, University of Chicago, Chicago, IL, United States of America, 5 Office of Laboratory Animal Resources, University of Colorado Anschutz Medical Campus, Aurora, CO, United States of America, 6 Animal Models Shared Resources, Georgetown University Medical Center, Washington, DC, United States of America, 7 Division of Animal Resources, Emory University, Atlanta, GA, United States of America, 8 Department of Comparative Medicine, School of Medicine, University of Washington, Seattle, WA, United States of America, 9 Comparative Medicine, Vertex Pharmaceuticals Incorporated, Boston, Massachusetts, United States of America, 10 Department of Psychiatry and Behavioral Sciences (By Courtesy), Stanford University, Stanford, CA, United States of America

* meglafollette@3rc.org

## Abstract

### Introduction

Despite increasing evidence that Environmental Health Monitoring (EHM) is an effective method to perform rodent colony health surveillance, promotes the 3Rs as a Replacement, is comparable or reduces cost, and demonstrates labor benefits, many research institutions continue to use live Soiled Bedding Sentinels (SBS). Some veterinarians and others responsible for rodent colony health monitoring remain cautious of the evidence supporting this new technology. Therefore, our objective was to perform a formal systematic review to identify, summarize, and evaluate the research on the efficacy of EHM as compared to SBS. This information is essential to establishing recommendations for rodent health monitoring programs.

### Methods

We systematically searched and evaluated all articles comparing EHM to SBS identified from PubMed, Web of Science, and CAB abstracts initially on November 12, 2021 with an update on Oct 15, 2023. Our inclusion criteria included publication in a peer-reviewed journal and collection of empirical data on rodent health monitoring alternatives to soiled bedding sentinels. Outcome data was extracted and analyzed via logistic regression and secondary descriptive statistics. Due to the nature of the included studies, no risk of bias assessment was performed. No specific funding was received for this review.

**Data Availability Statement:** All relevant data are within the manuscript and its Supporting Information files.

**Funding:** The author(s) received no specific funding for this work.

**Competing interests:** The authors have declared that no competing interests exist.

## Results

Forty-two peer-reviewed articles passed inclusion criteria for analysis. The design of studies varied substantially but included publications investigating exhaust dust testing (n = 27), sentinel-free soiled bedding (n = 8), and direct colony sampling (n = 24). Based on both logistical regression and descriptive criteria, all types of EHM appear to be superior to soiled bedding sentinels at detecting pathogens, regardless of their taxonomic classification.

## Conclusion

Based on these findings, we conclude there is a strong evidence base supporting the replacement of SBS with EHM. Looking forward, we encourage continued research on the detection of existing and emerging pathogens with these technologies and development of best practices for routine rodent health monitoring programs.

## Introduction

Rodent health monitoring programs detect unwanted pathogens in animal research facilities. As such they help ensure rodent health, personnel health, and the reproducibility and reliability of scientific research. Traditionally, rodent health monitoring programs have relied on sentinel rodents that are not directly involved with experimental studies. Rodent sentinels are exposed to pooled, soiled bedding from research animals in the colony. The rationale behind this is that the soiled bedding sentinel (SBS) animal becomes infected through exposure to a pathogen originating from the research colony. The disease is then detected in the sentinel animal by a variety of diagnostic assays or clinical signs. Typically, sentinel rodents have an intact immune system for pathogen detection by serology, but immunodeficient strains have also been used to better represent the population under surveillance or propagate the pathogen for easier detection by molecular methods. Sentinel animals are typically exposed at each cage-change, serially sampled, euthanized, and then replaced after 3–6 months. There is usually one SBS cage per rack, or per side of a rack, both to give specificity as to the location of a pathogen and to increase sensitivity by avoiding dilution of the pathogen in too large of a pool of soiled bedding. In any given research institution, hundreds or even thousands of rodents may be used each year exclusively for routine health surveillance without directly contributing to research data.

Despite SBS being standard practice for health monitoring programs, a 2016 systematic review of the SBS methodology could find only 15 relevant papers [1]. Systematic reviews follow a formal process to perform an exhaustive unbiased search of the literature, compile the results of individual papers into a meta-dataset, and then analyze this to detect consistent patterns of results [2]. De Bruin *et al.* [1] were able to find consistent evidence (in at least two studies) for the efficacy of SBS for mouse hepatitis virus (MHV), mouse parvovirus (MPV), Theiler murine encephalomyelitis virus (TMEV), *Helicobacter* spp., and fur mites; and evidence for a lack of efficacy for Sendai virus. However, for several other pathogens, there were either inconsistent results or evaluation by only a single suitable study [1]. From a cost- benefit and 3Rs perspective, the lack of supporting literature for SBS is concerning.

In recent years, Environmental Health Monitoring (EHM) has emerged as an alternative to SBS based health monitoring [3–10]. The rationale behind EHM is to bypass the use of sentinel animals and detect pathogens in the environment directly by monitoring for the presence of

**Table 1. Common terminology for environmental health monitoring.**

| | |
|---|---|
| Environmental Health Monitoring (EHM) | A variety of diagnostic sampling methods used to indirectly or directly perform rodent colony health surveillance without the use of sentinel animals. |
| Exhaust Dust Testing (EDT) | The use of swabs or a media to collect dust and nucleic acid that accumulates in **primary housing equipment** such as an individually ventilated cage (IVC) exhaust system. Most commonly associated with exhaust air dust (EAD™) testing or exhaust debris testing (EDx) of housing systems lacking cage level filtration where the sample is representative of an entire housing unit such as a rack of IVC cages. |
| EDT: Plenum Swabbing | The use of swabs (flocked or sticky) to collect dust and nucleic acid from specific locations within the exhaust system of an IVC rack. This sample represents a snapshot in time based on the material that is present at collection. |
| EDT: In-line Media | The positioning of media in the exhaust air current which collects dust and nucleic acid through air turbulence. The media continuously collects material thus representing the entire duration of the sampling period. |
| Sentinel-Free Soiled Bedding (SFSB) | The serial pooling of soiled bedding from rodent colony cages which is then sampled by either swabs and/or media for particulates and nucleic acid. Procedurally resembles a SBS program and commonly used where static caging or IVC with cage level filtration of exhaust air occurs. |
| SFSB: Indwelling Media/Swabs | The pre-placement of swabs or media within a soiled bedding container which is then exposed to particulates and nucleic acid during the entire sampling period. |
| SFSB: Indwelling: Agitation | The intermittent shaking or stirring of pooled soiled bedding to help expose indwelling swabs or media to particulates and nucleic acid. |
| SFSB: Indwelling: Stagnant | The absence of any dedicated agitation or stirring action of pooled soil bedding to expose indwelling media or swabs to particulates and nucleic acid other than what occurs through routine handling and serial addition of material. |
| SFSB: Single Exposure: Dredging | Use of swabs and/or media that are dredged through serially pooled soiled bedding at the end of the collection period for a single exposure to particulates and nucleic acid. |
| Research Equipment Monitoring (REM) | The use of swabs or media to sample husbandry or facility support equipment where dust and nucleic acid accumulate. Typically, the site of collection shows less precision in localizing the source and represents the health status of the animals within an entire housing room or facility. |
| Direct Colony Sampling (DCS) | The collection of feces, fur swabs, oral swabs, or blood to non-invasively (or minimally invasive) sample specific animals or their cage microenvironment. Typically used for quarantine testing of new arrival rodents or secondary/tertiary testing after a positive test by a different EHM method. This method is used due to the higher level of precision in pathogen localization at the cage level. |

their genetic material. In this context, the term "environmental" refers to space in the macro-environment of the housing room or within the mechanics of the caging system or micro-environment of the intra-cage space directly around the animal. Instead of live animals, EHM uses sterile swabs, flat media, or filter material to collect diagnostic samples within the animal cage, husbandry equipment, or soiled bedding. From these samples, the presence of pathogens in the environment is then detected by sensitive and specific PCR assays. A wide variety of different protocols for sample collection exist (see Table 1), but generally, they fall into 4 main approaches: Exhaust Dust Testing (EDT) which collects samples from the primary housing equipment; Sentinel-Free Soiled Bedding (SFSB) which collects samples from pooled soiled bedding; Room and Equipment Monitoring (REM) which collects samples from the room or husbandry equipment; and Direct Colony Sampling (DCS) which collects samples directly from colony animals themselves by non-invasive or minimally invasive methods.

EHM, in theory, has three distinct advantages over SBS. First, it avoids biological complications inherent to detecting the immune response of soiled bedding exposed sentinels as proxy for pathogen presence. More specifically, the mode of transmission for a variety of commonly excluded rodent pathogens is not through the fecal-oral route [11–13], and there are host-

adapted and environmentally labile pathogens not readily transmitted by soiled bedding [14]. Furthermore, sentinel susceptibility to infection may vary by animal age and strain [15–20]. Additionally, when testing is reliant on the occurrence of infection and seroconversion, the time to seroconvert can result in significant delays in detection [21]. [14] Second, because the sensitivity of EHM comes from the sensitivity and specificity of PCR based assays, EHM avoids the complexity of using differing methods of detection for different pathogens and can detect even minute levels of pathogen genomic sequences in samples collected from the environment [14]. Third, EHM eliminates any animal welfare concerns as well as removes animal costs and decreases labor costs inherent to a SBS program [22]. Thus, EHM is a potentially powerful 3Rs Replacement and can also benefit facility operations and overhead costs [22, 23].

Despite the advantages of EHM, EHM remains largely unadopted. In 2017, 95% of institutions were using SBS for rodent health monitoring [24]. Still, as of 2021, 89% of surveyed institutions were continuing to use SBS [22]. In the latter survey, 16% of participants believed EHM to have lower accuracy than SBS and 40% believed it to be difficult to use due to their caging types. Thus, a key roadblock to wider implementation is the perception of EHM as an inferior method compared to SBS, despite literature to the contrary.

As illustrated by de Bruin et al.'s work on SBS programs [1], systematic reviews are especially important when the scientific community does not have consensus on evidence-based best practices. First, they summarize the entirety of scientific literature on a particular topic in one place and subsequently expose decision-makers to evidence they may not have otherwise identified. Second, they counter confirmation bias (i.e., the subconscious tendency to pay more attention to evidence supporting a position and to discount evidence that refutes it). Third, they expose methodological issues and establish best practices in a particular research arm for the future.

Accordingly, here we present a systematic review of the evidence-base comparing the efficacy of EHM as compared to SBS. We had 4 aims that we aspired to achieve with this work. First, we formally compared the sensitivity of EHM to SBS across the current literature. Second, we formally tested whether any EHM methodologies were superior across all pathogens. Third, we formally tested whether any methods were better for detecting specific pathogens. Fourth, we provide an exhaustive catalog and qualitative summaries of the existing literature. In brief, this work shows that existing literature overwhelmingly supports EHM.

## Materials & methods

### Protocol

Prior to conducting the systematic review, we consulted the preferred reporting guidelines for systematic reviews and meta-analysis (PRISMA) guidelines, the Systematic Review Centre for Laboratory Animal Experimentation (SYRCLE) guides, and a veterinary science librarian. A completed PRISMA checklist can be found in **S1 File**. Our procedures were defined *a priori* in a study review protocol where we specified the search strategy, inclusion and exclusion criteria, and key data extraction items **S2 File.** We note that this file was submitted to the SYRCLE database after it had closed so the study cannot be considered formally registered. Additionally, we note the following changes to the protocol: conference proceedings were not searched to prevent duplication, no risk of bias was performed (see below for rationale), multiple reviewers extracted data, and discrepancy was resolved via discussion.

### Search strategy and article identification

Articles were identified first on November 12, 2021 and then on October 15, 2023 by searching PubMed, Web of Science, and CAB Abstracts (via CABI). We searched through titles,

abstracts, and keywords. Search terms included a term related to environmental health monitoring or exhaust air dust. Language was restricted to English. The full search strategy can be found in **S3 File**. Search records were uploaded to Covidence, de-duplicated, and screened.

## Eligibility criteria and article selection

After all duplicate articles were removed, all articles were initially screened by a single reviewer (CSC) for inclusion based on title and abstract. A random 20% of articles were independently assessed for inclusion by a second reviewer. Any disagreements were resolved via consensus. No automation tools were used. Prior to data extraction, full articles were additionally screened. The following inclusion criteria were used: (1) publication in a peer-reviewed journal; (2) published in English; (3) collection of original data; (4) with laboratory rats or mice; (5) assessing at least one environmental health monitoring technique. Studies were included even if they did not compare the efficacy of EHM to SBS directly. Exclusion criteria included: (1) reviews, (2) other species, (3) no use of EHM, (4) no relevant outcomes measure, and (5) not in English.

## Data extraction

**Sampling methodology.** For this review and to facilitate conversations in the field, common terminology was created to represent the variety of current methods constituting EHM (**Table 1**). EHM programs can further be broken down into 4 categories including: exhaust dust testing (EDT), sentinel-free soiled bedding (SFSB), research equipment monitoring (REM), and direct colony sampling (DCS). Several of these categories have been broken down further based on specific sampling methods. For each category, one individual (CSC) with an additional individual confirming (MRL) extracted the primary types of sampling methods that were evaluated in the publication.

In addition to the broad sampling types, one individual (CSC) also extracted potentially relevant information related to sampling methods including sampling material used (Media [including filter paper, filter media, contact media, gauze, etc.], Non-Adhesive Swabs (including cotton swabs, flocked swabs, and dry swabs), and Adhesive Swabs [i.e., Sticky Swabs]), SFSB sampling type (Agitation, Stagnant Exposure, or Dredging), length of time for media exposure, IVC rack sanitation details, total length of monitoring period, type of caging, IVC air changes per hour, frequency of health monitoring, mouse strain and sex, and bedding type. We also used three categories to describe information about EHM sampling material.

## Pathogen detection

We extracted which pathogens were tested for and designated whether they were detected with a (yes/no). A pathogen was considered to be detected if the sampling method picked up pathogen nucleic acid at least some of the time (for example 1 of 4 sentinel animals or 25% of media). Two coders per paper (CSC and a veterinarian) determined whether each sampling type detected (at least some of the time) each individual pathogen. Any discrepancies were resolved via discussion.

Pathogens were then recoded in three phases as shown in **S2 Table**. Coding categories were determined and confirmed by the laboratory animal veterinary co-authors (KRL, CAM, WHH, PLF, CPB, CBW). In the first phase, pathogens were grouped into specific category such as grouping *Helicobacter bills*, *Helicobacter ganmani*, *Helicobcter* spp., and *Helicobacter typhlonius* into a single *Helicobacter* spp. group. In the second phase, pathogens were lumped according to specific type such as grouping fur mites, *M. musculinus*, *Myobia musculi*, and *Radfordia affinis* into Ectoparasites. In the third phase, pathogens were grouped via two

methods. For method one, pathogens were coded based on type into the following categories: bacteria, fungus, parasite, or virus. For method two, pathogens were coded based on the agent's relative importance to the lab animal community which was modeled after the FELASA working group on revision of guidelines for health monitoring of rodents and rabbits that details which pathogens are recommended to be tested for on an annual or 3-month basis *versus* pathogens that are optional to be tested [25].

## Research coding

In addition to the above information, one individual (CSC) also extracted data based on the first author, publication year, journal name, country, title, terminology, number of cages, animals replaced, and any notes on money or time saved. Any missing information was left blank during data extraction and subsequently not included in summary statistics.

## Reporting quality & risk of bias assessment

Although due to the nature of the experimental designs of these studies, which included studies with no control conditions or animals, the ability to conduct a formal risk of bias assessment was limited, a modified version of the SYRCLE risk of bias assessment for animal studies was conducted by a single researcher (MRL). Results can be found in **S2 Table**. As neither animals nor cages are typically assigned to different treatment groups, question 1, 3, and 4 were removed from the tool; question 5 was also removed as interventions are not applied to a particular animal or cage. Additionally, for studies that only evaluated a single health monitoring method, were a cost analysis, or case study, all questions were considered not applicable.

## Data analysis

Data visualization and final preparation was performed in JMP Pro 16 for Windows and exported to SAS 9.4 for Windows for model selection and analysis. Given the differences in methodology and frequent lack of details regarding sample size, it was not possible to perform a formal weighted meta-analysis [26]. Instead, as described above, for each article for each combination of pathogen and detection method, we scored whether or not the method successfully detected the pathogen. The resulting data were analyzed as a logistic regression using SAS PROC GENMOD to implement a suitable generalized linear model. The dataset used for analysis and SAS code can be found in **S1 Dataset**. One of the advantages of this approach is that it is robust to missing data. Thus, studies which do not include SBS as a control, or do not compare all EHM methods, or all pathogens, can still be included and contribute appropriately to figuring the average efficacies of each factor combination. Note that data from Eichner and Smith 2023, while included in the descriptive statistics (i.e., total number of articles, pathogens evaluated, EHM methods use, etc.), was not included in formal data analysis due to significant published concerns with the experimental design and data collection [27, 28].

Our initial model was blocked by Article. For hypothesis testing, Sampling Method was nested within EHM vs SBS. This nesting approach implements a Hierarchical Linear Model, which explicitly tests whether SBS differs from EHM on average, and then whether the different EHM methods differ from each other, without the need for additional post-hoc tests. This approach is more powerful than comparing all 4 methods directly and also avoids p-hacking and HARKing by limiting the hypotheses to those defined ahead of time. We ran two versions of this analysis, where pathogen was coded by Type or by Importance (as described above). The interaction of EHM vs SBS by Pathogen Type or Importance (respectively, depending on analysis) was included to test for differences in the efficacy of Sampling Method for different Pathogens.

REML solutions suffer when model fit is too accurate, resulting in complete or quasi-complete separation of the model solution (i.e., the parameter estimates of the underlying regression model). When this happens, the parameter estimates expand with each iteration of the REML algorithm, and never reach a stable solution. This can be due to model overspecification, where unique combinations of variables perfectly predict outcomes or when a hypothesis-testing factor is too powerful in terms of predicting outcomes. We followed best practice in resolving these issues [29]. Overspecification is typically due to having too many categories within different variables. For experimental variables, such as specific pathogen, overspecification was solved by recoding the data into functional categories as described above. For blocking variables (like Article), which cannot be further condensed, these variables can be retained [29].

The initial model showed complete separation and could not generate a stable solution. This was due to including Articles which acted as a blocking factor, as most articles were missing many combinations of Sampling Method and Pathogen Type or Importance. Accordingly, we removed Article from the analysis following best practices for resolving these issues [29]. This yielded a quasi-complete solution with inflated estimates and errors. The interaction of EHM vs SBS by Pathogen Type or Importance was non-significant in either model and was removed. Once removed, the final model yielded a stable solution. The least square means and standard errors generated from the final models were subsequently graphed.

Having identified our final model, we then stress-tested the robustness of our findings by adopting alternative approaches, and trying to "break" the result. This approach is an important prophylactic to p-hacking. It is more recently referred to as "sensitivity analysis" [30] but in reality merely reflects long-standing best-practice in linear model design in whole-animal biostatistics [31]. To do so, we addressed two potential issues with our approach.

First, with respect to removing Article from the model, an alternative approach to the best practice recommended in [29], would be to treat Article as if it was a subject in a repeated measures analysis (i.e. as a random effect). While this approach is common in meta-analysis of continuous outcome variables [2], implementation in a logistic regression is complex. Most importantly, doing so involves Generalized Estimating Equations, which are sensitive to underlying assumptions and implementation, and often do provide consistent results across different software packages [32]. Furthermore, treating Article as a random effect is inherently questionable, not least because random effects pertain to a much larger unsampled population (of articles), whereas a systematic review aims to find the entire population of relevant articles. Despite these reservations, we performed this alternative analysis (see options in **S1 Dataset**). The results were unchanged, and so we present the more traditional best practice analysis described above.

Second, we initially included all articles, even if they did not include a SBS control, allowing the analysis to correct for these missing data. Accordingly, we reanalyzed the data using only articles which directly compared SBS and EHM, including stress-testing with the General Estimating Equations alternative approach described above. These analyses can be found in **S2 Dataset**. The results were unchanged, and so we present the analyses from the full data here.

**Descriptive analysis.** In addition to the logistic regression performed, we performed several descriptive analyses, tabulation, and visual display to synthesize the results. The datasets used for these analyses can be found in **S1 Table** which reports a detailed list on pathogens per article and **S2 Table** which present relevant article level data. These allow for a more granular view of the data on a per pathogen basis that is not possible via logistic regression. We present summarized counts of the number of articles that evaluated each pathogen and the number that successfully detected each pathogen. Additionally, we present counts of various study characteristics.

## Results

### Study selection

As shown in the PRISMA diagram in **Fig 1,** our literature search resulted in 862 peer-reviewed publications. After excluding duplicates, a total of 775 abstracts were screened; a full list of these studies with reasons for exclusion can be found in S3 Table. After meeting inclusion criteria based on abstracts, a total of 53 full-text papers were subjected to full-text screening. Ultimately, based on our inclusion criteria, the final sample resulted in 42 articles (5.42% of the total initial pool after duplicates were removed) published between 2004 and 2023. There was an international representation of researchers including corresponding authors from North America (n = 27), Europe (n = 9), Asia (n = 5), and South America (n = 1). The articles were published primarily in the Journal of the American Association for Laboratory Animal Science (n = 23).

### Efficacy of sampling method and pathogen

The extracted data were analyzed by Pathogen recoded as Pathogen Type or Pathogen Importance. When Pathogens were coded by Pathogen Type, the Pathogen Type as it related to

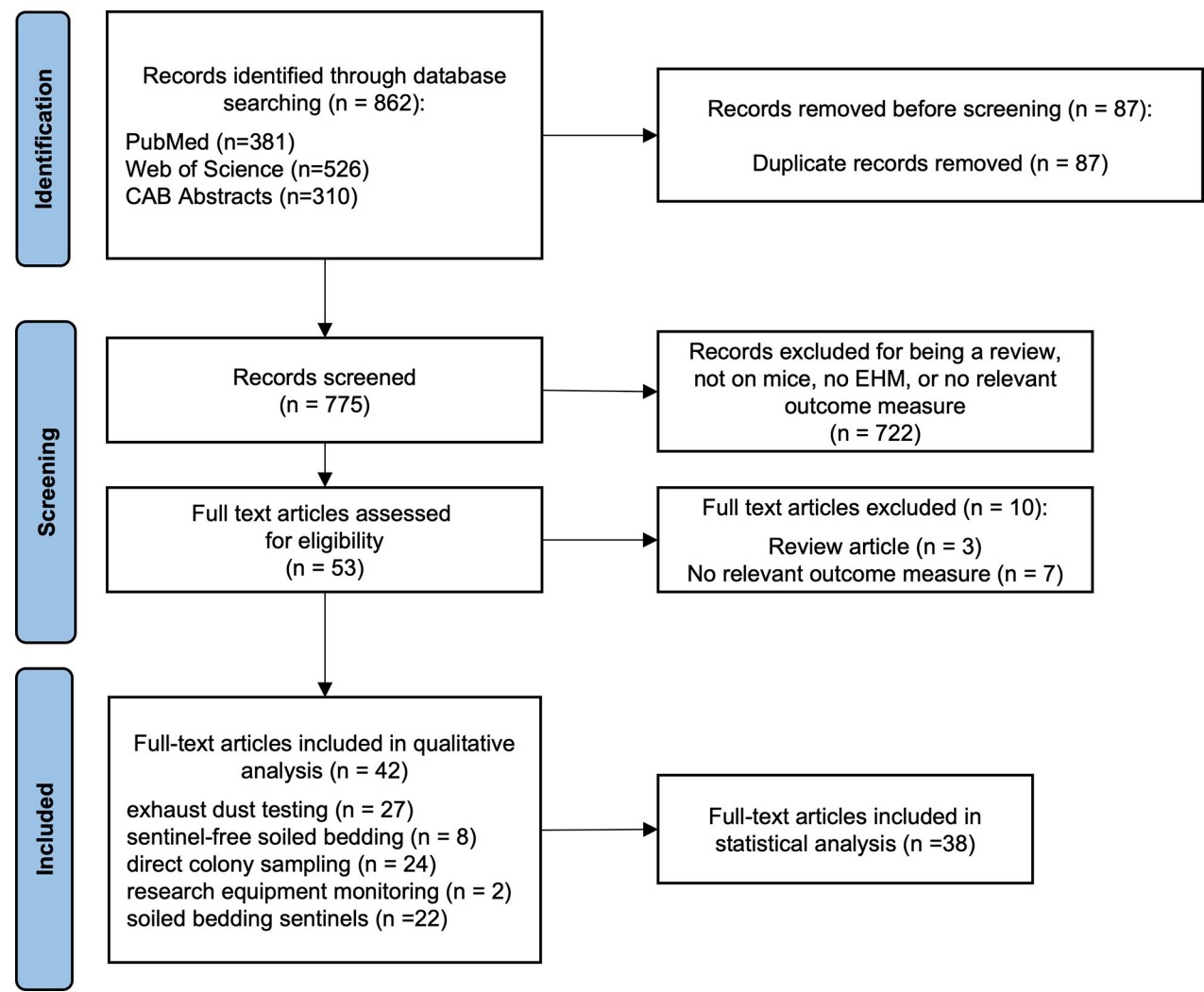

**Fig 1. Preferred Reporting Items for Systematic Reviews and meta-Analyses (PRISMA) flow diagram.**

# **Environmental health monitoring** detects pathogens more often than soiled bedding sentinels, regardless of sampling method or pathogen type.

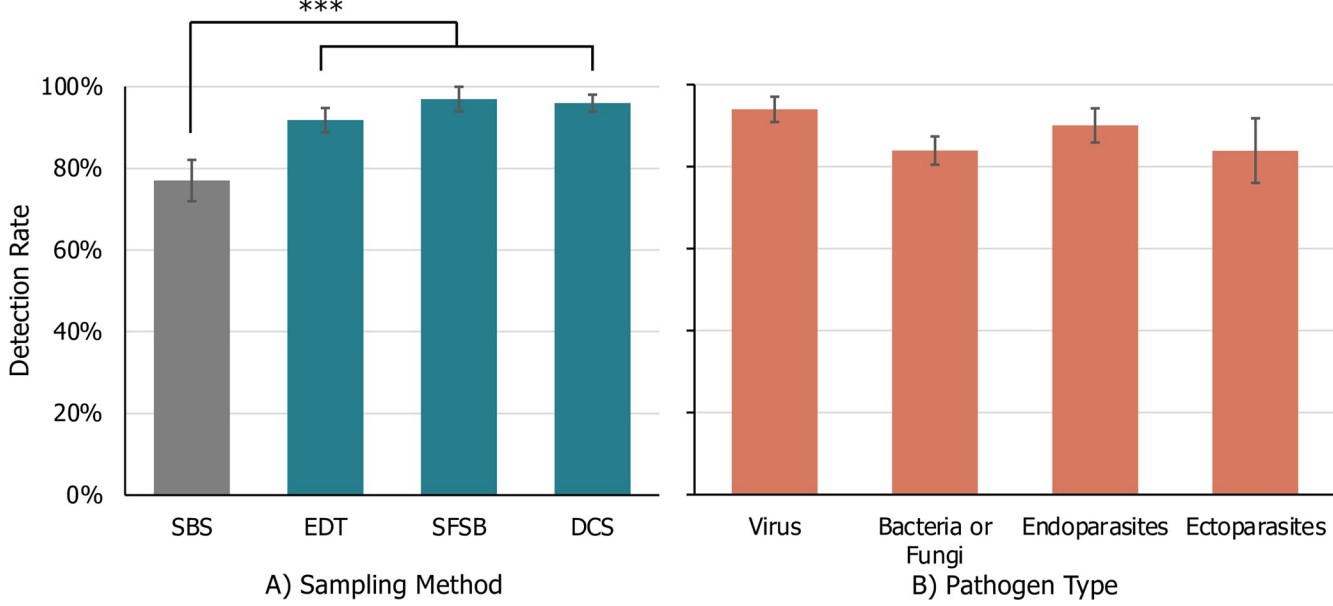

**Fig 2. Comparing pathogen detection by type between environmental health monitoring vs soiled bedding sampling.** The detection rate (LSM +/- SE) of A) each rodent health monitoring strategy (soiled bedding sampling (SBS), exhaust dust testing (EDT), sentinel-free soiled bedding (SFSB), or direct colony sampling (DCS)) across B) pathogens coded by type (bacteria/fungus, ectoparasites, endoparasites, viruses) as investigated in 30 articles. N = 252, ***p < 0.0001.

Sampling Method interaction was non-significant and caused quasi-complete separation of the model and was thus removed. Across sampling methods, detection rate did not differ by Pathogen Type (LR Chi-Sq = 3.34; P = 0.3422; **Fig 2**). Detection rate was significantly higher for EHM methods overall vs SBS (LR Chi-Sq = 18.60; P<0.0001; **Fig 2**). The EHM methods did not differ significantly amongst themselves (LR Chi-Sq = 2.04; P = 0.3597; **Fig 2**).

When Pathogens were coded by Importance, the Pathogen Importance by Sampling Method interaction was non-significant and caused quasi-complete separation of the model and was thus removed. Across sampling methods, detection rate was higher for pathogens tested on a 3-Month or Annual basis as compared to Optional tested (LR Chi-Sq = 5.33; P = 0.0210; **Fig 3**). Detection rate was significantly higher for EHM methods overall vs SBS (LR Chi-Sq = 18.90; P<0.0001; **Fig 3**). The EHM methods did not differ significantly amongst themselves (LR Chi-Sq = 2.18; P = 0.3358; **Fig 3**).

## **Pathogen-level descriptive analysis**

In addition to a formal statistical analysis, we also present the data on a per pathogen basis which is summarized in **Table 2** with detailed data on a per pathogen basis in **S1 Table**. Overall, the studies evaluated 26 different pathogens. Looking only at experiments that directly compared SBS to EHM, SBS failed to detect pathogens in 29% cases where EHM detected a pathogen, while EHM failed only in 3% of cases where SBS detected a pathogen (**Fig 4**). This comparison includes 12 different publications, where there were 20 instances where SBS failed to detect 11 different pathogen sof interest and EHM succeeded (**Fig 4** **and Table 2**).

Across all studies, we considered a health monitoring technique to be effective if it detected the pathogen at least some of the time by at least two articles, consistent with the criteria

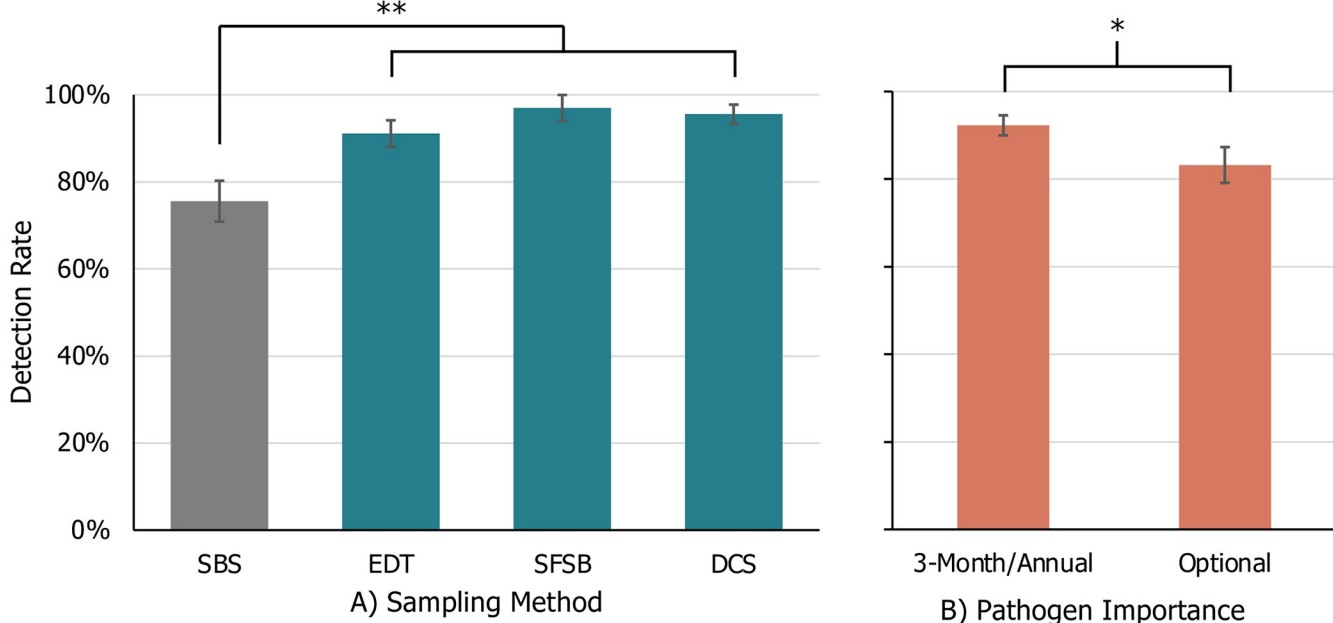

**Fig 3. Comparing pathogen detection by importance between environmental health monitoring vs soiled bedding sampling.** The detection rate (LSM +/- SE) of A) each rodent health monitoring strategy (soiled bedding sampling (SBS), exhaust dust testing (EDT), sentinel-free soiled bedding (SFSB), or direct colony sampling (DCS)) across B) pathogens coded by "importance" (per FELASA guidelines as to whether they should be tested on a 3-month/annual basis or optionally) as investigated in 30 articles. N = 252, **p < 0.001 *p<0.025.

established by de Bruin et al. Based on this standard, EDT and SFSB appear to be effective at detecting 10 pathogens including mouse norovirus (MNV), mouse parvovirus (MPV), *Helicobacter* spp., *Proteus mirabilis*, *Rodentibacter* spp., *Staphylococcus* spp., Ectoparasites, *Entamoeba* spp., *Spironucleus* spp., and pinworms (*Aspiculuris* spp. & *Sypacia obvelata*). Additionally, EDT appears to be effective for 6 additional pathogens including mouse hepatitis virus (MHV), murine astrovirus, *Tritrichomonas* spp., *Klebsiella* spp., *Corynebacterium bovis*, and *Pneumocystis* spp, and Tritrichomonas spp. Finally, several other pathogens were only evaluated in a single paper as shown in **Table 2** and therefore no broad conclusions were drawn.

## Study characteristics

The detailed characteristics of the 42 studies included are present in **S2 Table** with a high-level summary provided in **Table 3.** The design of studies varied substantially including 38 experimental studies and 5 case studies. It included publications investigating EDT (n = 27), SFSB (n = 8), and DCS (n = 24). Studies were split as to whether they investigated multiple EHM methods (n = 17), a single EHM method (n = 25), and whether they compared at least one EHM method to SBS (n = 22). As mentioned above, the statistical methods used are robust with respect to missing data, are well suited to this heterogeneity, and were extensively stress-tested regardless. The studies included three articles that did not have data suitable for quantitative statistical analysis (one article was a cost analysis, one was a protocol, and one was the sole article detailing REM).

Of the articles using EDT and SFSB, a variety of materials and methods were used to collect samples. For EDT, 14 studies used more than one sampling material type. These sampling

**Table 2. Efficacy of pathogen detection with Soiled Bedding Sentinels (SBS), Exhaust Dust Testing (EDT), Sentinel-Free Soiled Bedding (SFSB), and Direct Colony Sampling (DCS).**

| Viral Agents | SBS | EDT | SFSB | DCS | EHM Detected Where SBS Failed? |
|---|---|---|---|---|---|
| Lactate Dehydrogenase Elevating Virus | | 1/1 | | | |
| Lymphocytic choriomeningitis virus | | 1/1 | | | |
| Murine Hepatitis Virus | 2/2 | 2/2 | 1/1 | 2/2 | |
| Murine Norovirus | 8/8 | 5/6 | 4/4 | 4/4 | |
| Murine Parvoviruses | 3/3 | 4/4 | 3/3 | 4/4 | |
| Murine Astrovirus | 3/3 | 2/2 | 1/1 | 2/2 | |
| Murine Rotavirus | 0/1 | 0/1 | | | |
| Sendai Virus | 0/1 | 1/1 | | | Yes (1) |
| Theiler Murine Encephalomyelitis Virus | 1/1 | | 1/1 | | |
| **Bacterial Agents** | **SBS** | **EDT** | **SFSB** | **DCS** | |
| *Chlamydia muridarum* | 1/1 | 1/1 | | 2/2 | |
| *Citrobacter rodentium* | 1/1 | 1/1 | | 2/2 | |
| *Corynebacterium bovis* | | 2/2 | | | |
| *Helicobacter* spp. | 8/10 | 10/10 | 4/4 | 10/10 | Yes (2) |
| *Klebsiella* spp. | 2/2 | 2/3 | 1/1 | 2/3 | |
| *Mycoplasma* spp. | 1/1 | 1/1 | | 1/1 | |
| *Proteus mirabilis* | 2/5 | 4/4 | 2/2 | 3/4 | Yes (3) |
| *Pseudomonas aeruginosa* | 1/1 | 1/2 | | 1/1 | |
| *Rodentibacter* spp. | 5/8 | 10/10 | 2/2 | 8/8 | Yes (3) |
| *Staphylococcus* spp. | 6/7 | 6/7 | 2/2 | 6/7 | Yes (1) |
| *Streptococcus* spp. | 2/2 | 0/1 | 0/1 | 2/3 | |
| **Fungal Agents** | **SBS** | **EDT** | **SFSB** | **DCS** | |
| *Pneumocystis* spp. | 0/2 | 2/2 | | | Yes (2) |
| **Parasites** | **SBS** | **EDT** | **SFSB** | **DCS** | |
| Ectoparasites (*Myobia musculi*, *Myocoptes musculinus*, *Radfordia affinis*) | 3/7 | 9/9 | 3/3 | 8/8 | Yes (4) |
| *Entamoeba* spp. | 4/5 | 5/6 | 2/2 | 7/7 | Yes (1) |
| Pinworms (*Aspiculuris* spp. *Syphacia obvelata*) | 4/5 | 4/4 | 2/2 | 5/5 | Yes (1) |
| *Spironucleus* spp. | 2/3 | 2/2 | 2/2 | 4/4 | Yes (1) |
| *Tritichomonas* spp. | 3/4 | 5/5 | 1/1 | 5/5 | Yes (1) |
| **TOTAL** | **61/83 (73%)** | **81/88 (92%)** | **31/32 (97%)** | **78/82 (95%)** | **12 Articles** |

This data summarizes detailed data found in S1 Table, listing the number of articles showing that the sampling method detected the pathogen at least some of the time/ the total number of articles evaluating that sampling method for the specific pathogen. The far-right column notes the articles in which EHM detected a pathogen where SBS failed followed by the number of articles in paratheses.

materials included media (n = 22), non-adhesive swabs (n = 11), and adhesive swabs (n = 5). For SFSB, 5 studies used more than one sampling material type. These sampling materials included media (n = 6), non-adhesive swabs (n = 5), and adhesive swabs (n = 2). SFSB exposure methods included agitation (n = 5), dredging (n = 3), or stagnant exposure (n = 3).

All studies were performed with mice. For mouse characteristics, 18 studies used only female mice, 9 used both sexes, 3 used only males (11 did not report mouse sex and 1 was a cost analysis.) Additionally, 18 studies used a common outbred stock (Swiss Webster, CD1, or ICR), 5 used C57BL/6, 3 used nude (Foxn1[nu]/Foxn1[nu]) [ta] mice, and 16 used other mouse strains or stocks (various vendors). One study did not report mouse strain. Mouse cages

**In head-to-head studies, where environmental health monitoring detected a pathogen, soiled bedding sentinels failed in 29% of cases. Conversely, in head-to-head studies, where soiled bedding sentinels detected a pathogen, environmental health monitoring failed in 3% of cases.**

a)

|  | | SBS No | SBS Yes | SBS sensitivity vs EHM |
|---|---|---|---|---|
| EHM | No | 1 | 2 | |
| EHM | Yes | 20 | 65 | 60/85 (**71%**) |
| **EHM sensitivity vs SBS** | | | 65/67 (**97%**) | |

b)

| Pathogen Name | Detection by SBS/EHM |
|---|---|
| *Pneumocystis* spp. | 0/2 |
| Sendai Virus | 0/1 |
| *Proteus mirabilis* | 2/5 |
| Ectoparasites | 3/7 |
| *Rodentibacter* spp. | 5/8 |
| *Spironucleus* spp. | 2/3 |
| *Tritrichomonas* spp. | 3/4 |
| *Entamoeba* spp. | 4/5 |
| Pinworms | 4/5 |
| *Helicobacter* spp. | 12/14 |
| *Staphylococcus* spp. | 6/7 |
| *Streptococcus* spp. | 2/1 |
| *Pseudomonas aeruginosa* | 1/0 |

**Fig 4. Head-to-Head comparison of EHM & SBS.** This figure summarizes detailed data found in S1 Table. It examines only head-to-head experiments which directly compared soiled bedding sentinels (SBS) to environmental health monitoring (EHM). **a**) Confusion matrix summarizing these experiments, given the differing methodologies involved. Thus, when EHM detected a pathogen, SBS failed in 29% of these cases (i.e., SBS showed a sensitivity of 71% relative to EHM), see EHM "Yes" row. Conversely, when SBS detected a pathogen, EHM failed in 3% of cases (i.e., EHM showed a sensitivity of 97% relative to SBS), see SBS "Yes" column. **b**) Breakdown, by the pathogens evaluated, where experiments found differing results between EHM and SBS (pathogens with perfect agreement are excluded). The number of these experiments where SBS detected the pathogen / the number of where EHM detected the pathogen is shown in order of relative SBS performance.

contained bedding made of corncob (n = 17), wood (n = 17), paper (n = 2), or wood/paper (n = 1), when reported. All but four articles used IVCs to house the animals (those used static caging). Of the IVCs, most used rack-level filtration (n = 30) and some cage-level filtration (n = 8). The number of cages used within each article ranged from 2 to 21,000, with an average of 1,814 cages (SD = 5766).

Cage or rack ventilation and sanitation also varied greatly and were not always reported. Most often, 60 to 70 IVC air changes per hour were reported (n = 15), 30 to 45 air changes per hour were also sometimes used (n = 4), and 8 reported other values. Rack sanitation, when reported, was often achieved using a rack washer (n = 14) and/or autoclaving (n = 14). Cage sanitation, when reported, was often achieved with a mechanical washer (n = 9) and/or autoclaving (n = 19).

## Terminology

The included articles used a wide variety of terms to describe rodent health monitoring and environmental health monitoring as distinct concepts. All 42 articles used the term "monitoring," 38 articles used the word "health," and 34 used the compound term "health monitoring." Additional common terms included "microbiological" or "microbiota" monitoring (n = 19),

**Table 3. Specific methods used for health monitoring.**

| Ref # | First Author | Year | Study Type | Health Monitoring Evaluated | Length of monitoring (weeks) | Ventilation and Filtration Method | Sex of Mice | Bedding Type |
|---|---|---|---|---|---|---|---|---|
| [33] | Bauer | 2016 | E | EDT, DCS | 12 | Rack | ? | Paper |
| [34] | Buchheister | 2020 | E | EDT, SBS | ? | Cage, Static | F | Wood |
| [35] | Butt | 2023 | E | DCS | 16 | Rack | ? | Wood |
| [36] | Chou | 2020 | E | DCS | N/A | ? | M, F | ? |
| [37] | Clancy | 2022 | C | EDT, DCS | 36 | Rack | ? | Corncob |
| [12] | Compton | 2004 | E | EDT, SBS | 10 | Cage | F | Corncob |
| [38] | Compton | 2015 | E | REM | 12 | Rack | F | Corncob |
| [39] | Compton | 2017 | E | EDT, DCS, SBS | 4 | Rack | F | Corncob |
| [4] | Dubelko | 2018 | E | SFSB, DCS, SBS | 12 | Cage | F | Corncob |
| [27] | Eichner | 2023 | E | SFSB, SBS | 24 | Cage | M, F | Wood |
| [40] | Gerwin | 2017 | E | SFSB, SBS | 12 | Cage | F | Wood |
| [5] | Hanson | 2021 | E | SFSB, SBS | 12 | Cage, Static | F | Corncob |
| [41] | Iwantschenko | 2022 | E | EDT, SFSB, DCS | ? | Rack | F | Wood |
| [42] | Jacobsen | 2005 | E | DCS | 20 | Cage | M, F | Wood |
| [3] | Jensen | 2013 | E | EDT, DCS | 10 | Rack | M, F | Wood, Paper |
| [43] | Kapoor | 2017 | E | EDT, DCS | 13 | Rack | M | Corncob |
| [44] | Kim | 2023 | E | EDT, DCS | 12 | Rack | F | Wood |
| [45] | Korner | 2019 | E | EDT, SBS | 12 | Rack | M, F | Wood |
| [46] | Leblanc | 2014 | E & C | EDT, DCS, SBS | Unclear | Rack | F | Corncob |
| [47] | Luchins | 2020a | C | EDT | N/A | Rack | M,F | Corncob |
| [23] | Luchins | 2020b | E | EDT, SBS | 52 | Rack | F | Corncob |
| [48] | Lupini | 2022 | E | EDT, DCS | 4 | Rack | ? | ? |
| [6] | Mahabir | 2019 | E | EDT, DCS | 12 | Rack | M | ? |
| [7] | Mailhiot | 2020 | E | EDT, SBS | 52 | Rack | F | Corncob |
| [8] | Manuel | 2016 | E | EDT | 2 | Rack | F | Wood |
| [49] | Manuel | 2017 | C | EDT | 8 | Rack | M | Wood |
| [50] | Miller | 2016 | E | EDT | 3 | Rack | F | Wood |
| [51] | Miller | 2016 | E | EDT, SBS | 12 | Rack | M, F | Wood |
| [52] | Miller | 2018 | E | EDT, SBS | 12 | Rack | F | Wood |
| [53] | Niimi | 2018 | E | EDT, DCS, SBS | 12 | Rack | ? | Wood |
| [9] | O'Connell | 2021 | E | SFSB, SBS | 8 | Rack | F | Corncob |
| [54] | Parkinson | 2011 | C | DCS, REM | N/A | N/A | N/A | ? |
| [10] | Pettan-Brewer | 2020 | E | EDT, DCS | 16 | Rack | ? | ? |
| [55] | Ragland | 2019 | E | EDT, DCS | 52 | Rack | ? | Paper |
| [56] | Scavizzi | 2021 | E | DCS, SBS | 12 | Rack | M, F | Corncob |
| [57] | Schlapp | 2018 | E | EDT, SBS | 16–24 | Rack | F | ? |
| [58] | Srinivasan | 2021 | E | DCS | ? | Static | ? | Corncob |
| [59] | Srinivasan | 2022 | E | DCS | ? | Static | ? | Corncob |
| [60] | Tierce | 2022 | E | DSC, SBS | 10 | Rack | ? | Corncob |
| [61] | Varela | 2022 | E | EDT, SFSB, DCS, SBS | 13 | Rack | ? | Corncob |
| [62] | Winn | 2022 | E | SFSB, DCS, SBS | 12 | Cage | F | Wood |
| [63] | Zorn | 2016 | E | EDT, SBS | 12 | Rack | M, F | Wood |

The below table lists key characteristics of the specific methods used in the 42 articles included in the analysis. Type indicated the type of study (E = Experimental Research, C = Case Study), health monitoring methods evaluated (SBS = Soiled Bedding Sentinels, EDT = Exhaust Dust Testing, SFSB = Sentinel Free Soiled Bedding, REM = Research Equipment Monitoring, DCS = Direct Colony Sampling), total duration of monitoring method, ventilation and filtration type (Rack = IVC Rack-level filtration, Cage = IVC Cage-level filtration, Static), the sex of animals used within the study, and the type of bedding. When articles had 2 experiments with different values those values are split by a comma. "?" = not reported. Additional information on article characteristics can be found in **S2 Table**.

"hygiene" or "hygienic" (n = 12), "surveillance" (n = 14), or "routine" (n = 14). For environmental health monitoring, 30 articles used the term "environmental" at least once, while the other 13 only addressed monitoring strategies explicitly (i.e., "exhaust air monitoring"). Of the 30 articles that mentioned exhaust dust testing, 27 used the term "exhaust," 26 used the term "air," 20 used "dust," and 16 used "monitoring."

## Bias assessment

In the risk of bias assessment (results found in **S2 Table**), 12 articles were excluded due to only evaluating a single method for rodent health monitoring, being a case report, or a cost analysis. The following statements refer to only articles included in the bias assessment. Most articles had similar groups at baseline (n = 27), and appeared to be free of other problems that could result in bias (n = 28). Additionally, for most articles it was neither applicable for animals/cages to be selected at random (n = 28) nor did they have incomplete outcome data that needed to be addressed (n = 25). For most articles it was unclear if outcome assessors were blinded (n = 26). All articles appeared to be free of selective outcome reporting.

## Discussion

### Environmental health monitoring has superior pathogen detection over soiled bedding sentinels

Overall, the data from this systematic review shows that EHM has superior pathogen detection over that of SBS. This conclusion is supported by both logistic regression and descriptive statistics regardless of pathogen taxonomic classification (such as between viruses and ectoparasites), pathogen importance (such as between agents recommended to be tested every 3-months/annually versus optionally), or type of EHM (such as between EDT, SFSB, or DCM). Therefore, EHM has improved detection over SBS for the majority of agents on each institution's exclusion list and this holds true regardless of the classification and testing frequency. Across all of the articles identified and included in the systematic review, at least two peer-reviewed articles have shown that at least one type of EHM method can detect 18 different pathogens across taxonomic classifications. In particular, there is good evidence for EDT consistently detecting 16 pathogens and SFSB detecting 10 pathogens, again across taxonomic classification. When compared to the only systematic literature of SBS methods, SBS only demonstrated good evidence of detecting 5 pathogens [1].

The studies reviewed include investigations in different facilities, continents, bedding types, mouse strains, caging/rack types, and ventilation changes. Despite this diversity in their exact components and study designs, generally consistent conclusions were met in that EHM is superior to SBS. Still, we recommend that future research should accurately report all potentially relevant housing factors for reference.

Overall, there were 21 cases where SBS failed to detect a particular pathogen compared to 6, 1, and 3 cases where EDT, SFSB, and DCS failed to detect a particular pathogen, respectively. To follow, we will discuss the publications in which EHM methods failed to detect particular pathogens, as these studies are sometimes cited as rationale against switching to EHM. Some of these articles had higher risks of bias as indicated in S2 Table.

In Compton 2004, EDT failed to detect Murine Rotavirus [12]; however, in this case, soiled bedding sentinels also failed to detect this pathogen and therefore this finding cannot be used to demonstrate the superiority of SBS.

In Mahabir 2019, EDT failed to detect *Klebsiella* spp., *Staphylococcus* spp., and *Streptococcus* spp. [6]; however, in the discussion authors note that this was likely due to low prevalence

and/or organism numbers of *Staphylococcus* spp. (5–9 cages) and *Klebsiella* spp. (1 cage), as well as the absence of shedding of *Streptococcus* spp. In fact, they note that at the end of the study *Streptococcus* spp. was not detected in any mouse. Additionally, SBS was not evaluated so it's unknown if it would have detected these pathogens. We note that EDT has successfully been found to detect *Staphylococcus* spp. in 6 additional papers [44, 52, 53, 55, 57, 61] and *Klebsiella* spp. in 2 publications [52, 55].

Bauer 2016 is often cited as demonstrating that EHM is not effective as the paper concluded that EDT consistently failed to detect MNV (and only inconsistently detected MPV) [33]. At the time of publication, the authors did not have an explanation for this finding. However, the literature and methods behind EDT have evolved since then. A likely explanation is that early testing methods used in this paper were less sophisticated than current standards. In particular, the type of filter paper used was rudimentary and the size of the filter paper was smaller than modern materials which may have led to lack of detection. Additionally, this study did have SBS as a control condition and it is possible the infected mice were no longer shedding in the case of MPV. We note that MNV has successfully been detected by EDT in 5 additional publications [7, 10, 50, 52, 61].

In Miller 2018, EDT failed to detect *Pseudomonas aeruginosa* when SBS did detect this pathogen [52]. In this case, authors hypothesized this could be due to pooling, infected mice being removed before gauze pieces were installed (gauze were only present for 3 weeks), or a false positive in bacterial culture or a transient infection that did not lead to shedding. *Pseudomonas aeruginosa* was detected by EDT in Ragland 2019 [55].

Finally, in Winn 2022, SFSB and DCS failed to detect beta-haemolytic Streptococcus while SBS did detect it [62]; this is likely due to the prevalence of this opportunistic agent being below the sensitivity for the assay, which was indicated by low copy numbers attained via SBS. Additionally, in Winn 2022, DCS failed to detect *Klebsiella* spp. or *Proteus mirabilis*, again, likely due to lower organism copy numbers in that study; however, we note that SFSB was able to detect these pathogens, and DCS is generally recommended as a supplementary rodent health monitoring strategy.

A systematic review is only as good as the studies that are included in it, and due to the differences in testing parameters for animal versus replacement studies, the formal risk of bias assessment we performed was limited, although indicated relatively low risk of bias across studies. Furthermore, as the studies reviewed in this manuscript do not deal with determination of causality (and instead are focused on the ability of a technique to detect), there is much less risk of bias. There are no animals to assign to treatment groups or caretakers to blind to treatment allocation. The biggest risk for bias comes from publication bias (i.e., not publishing negative results). Given that the majority of the studies compared SBS to EHM, this is unlikely as both methods would have to fail to detect all pathogens, whereas a failure of EHM to detect a pathogen when SBS was successful would be a more alluring finding for publication. In the smaller group of papers that only examined EHM, the risk of publication bias is arguably higher. However, including or excluding these papers do not affect the outcome of our statistical analyses, arguing against any significant bias.

## Ethical, scientific, and practical implications

Rodent health monitoring programs are intended to assist research laboratories in detecting and/or determining absence of adventitious pathogens, but the reliability and success rate of methods used is now receiving more scrutiny. Despite SBS programs having been the traditional practice for many years, the 2016 systematic review on the efficacy of this methodology only found evidence of ability to detect 5 pathogens [1].

Conversely, the results of this review clearly demonstrate the superiority of EHM methods in pathogen detection. Therefore, we argue that there is a scientific and ethical imperative for programs to work towards replacing SBS with EHM. Full implementation of the 3Rs (Refinement, Reduction, and Replacement) is considered an essential part of conducting humane science. There were 4 papers included in this review that mentioned the specific number of animals that would be replaced at their facilities due to switching to EHM [7, 10, 23, 52]. Across just these four articles a total of 6,876 animals will be replaced yearly. In our recent survey, 111 institutions indicated the potential for a total of well over 20,000 additional rodents to be replaced each year [23].

Beyond the positive ethical and scientific implications of EHM, there was often frequent discussion of cost and time savings to the technique. Thirteen articles mentioned cost savings and this review includes one formal cost analysis [23]. An additional article indicated the technique reduced costs 3-fold [7]. Additionally, six articles mentioned time savings, indicating benefits from not needing to check sentinel mice, spending less time on cage changes/care and diagnostic testing procedures.

Of course, dismantling and replacing traditional health monitoring programs takes time and effort. Our recent cross-sectional survey indicates that although personnel generally believe EHM is good, they do not feel professional pressure to make the switch to EHM and are not confident in their ability to make the switch [22]. Additionally, all three factors are positively associated with intentions to switch to EHM. It is therefore imperative to create pressure to switch to EHM and provide resources to increase confidence in making the switch. To support this aim, the 3Rs Collaborative has created an extensive resource hub on the topic that includes basic information about EHM, lists names and logos of institutions that have switched, editable standard operating procedures, an editable slide deck for internal presentations to gain consensus for the switch, sanitation information, cost analysis information, frequently asked questions, information on how to switch, and records of prior presentations (www.3rc.org/ehm).

Although we recommend EHM as the core part of rodent health monitoring, it should not be considered a substitution for techniques required to detect and characterize novel agents. After all, as EHM is reliant on PCR it can miss any agents that are novel or not included in the testing panel. Therefore, if abnormal clinical signs or study results are observed in colony rodents without a corresponding diagnosis via the EHM panel, we recommend performing a thorough diagnostic workup (e.g., histopathology, clinical chemistry, blood count) on the affected colony animals.

## Concordance is needed on terminology for environmental health monitoring

Our findings in this article and general expertise in the field highlight the need for concordance on terminology for EHM. The success of EHM calls for a paradigm shift in the methods used for routine rodent health monitoring. To assist in this shift to EHM, clear, and accurate communication is needed. While conducting this systematic review, it was clear that there was a need for both new standardized terminology and an organizational structure for what constitutes EHM. At its core, and by our definition, EHM involves collecting a sample for routine health monitoring without the need for live sentinel animals.

After reviewing the literature and having consensus discussions on the frequency of terminology in articles, vendor-specific terminology, and technique-specific factors our expert working group created a table of preferred terminology (**Table 1**). Thus far, we divide EHM into 4 main types based on sample collection. These include exhaust dust testing (EDT) within

IVC racks, sampling pooled soiled bedding by a variety of techniques (SFSB), sampling dust from the housing room or husbandry equipment (REM), and finally the sampling of the intra-cage environment of individual cages, or minimally invasive sampling of individual research animals by direct colony sampling (DCS). For EDT and SFSB, various variations to the primary method are being investigated to increase detection sensitivity while still being practical. As EHM techniques are refined over time, we strongly encourage laboratory animal veterinarians and managers to consistently use these vendor-agnostic terms to advance knowledge more quickly and allow research to be identified quickly by other professionals.

## Future research–targets and best practices

As of October of 2023 (the end of the sampling period for this article), only 8 publications had reported outcomes from SFSB sampling. This is likely due to the lag in the development of this technique compared to EDT with reports of SFSB validation only appearing since 2017. So far, these publications have reported using a variety of sampling materials and methods (although most with success). Still, further research is needed to determine what specific materials and methods are most effective, especially since this method can be used on all caging/rack types.

In particular, there may be concern with respect to monitoring mice in specific opportunistic pathogen free and/or immunodeficient health status settings, especially where infection rates or prevalence may be low. Thus far, there is good support to demonstrate the ability of EDT to detect and help eliminate *Corynebacterium bovis* from immunodeficient mice within barrier rooms [8, 49] and that SFSB can detect murine astrovirus and other pathogens in specific pathogen free mice [62]. Continued investigations may be beneficial to demonstrate the utility of EHM in areas with extensive exclusion lists.

Independent of the health monitoring methods used, it would also be beneficial to conduct research on the effect of sample pooling on pathogen detection by PCR assays. Pooling refers to operating procedures where individual samples collected from a cohort (generally fecal pellets, oral swabs, or fur swabs) are combined and processed as one sample. For EDT and SFSB, pooling typically refers to combining media and/or bedding from multiple racks. Pooling presents several practical challenges. For instance, the volume of samples that can be combined into one test is physically limited; and/or sample homogeneity can suffer at greater pooling volumes [64]. Additionally, there is the possibility that pooled samples dilute the signal for low-prevalence infections. Further research in pooling samples by DCS as a direct comparison to or supplement with other environmental health monitoring methods is needed. Regardless, all such methods can continue to replace the use of sentinel animals.

For any future research on EHM, we recommend the following best practices. First, a gold standard control—such as a naïve contact sentinel—should be used within each study to demonstrate if the pathogen of interest is being shed and, more importantly, indicate whether the pathogen of interest is infectious and leading to infection at the peak of the shedding period [11, 18, 65, 66]. This is necessary as it is possible for infected mice to shed neutralized virus and direct testing of infected mice will indicate the presence of a virus even though it is no longer of a concern to the broader colony. Second, future studies should follow both the PRE-PARE guidelines [67] and ARRIVE guidelines [68] in reporting to help ensure good study quality and assist future systematic reviews. These guidelines emphasize essential experimental design concepts like randomization and blinding. For EHM studies in particular, authors should report relevant items such as type of bedding, air changes per hour, sanitation protocols, and type of caging/rack. Third, authors should use the recommended terminology used in this article to assist in future identification of articles.

## Limitations

One potential criticism of this systematic review is that we did not include conference posters or presentations about EHM. However, this is best practice for two reasons. First, we avoid the duplication of the same experiment in the data set (due to multiple abstracts at different conferences and/or final peer-reviewed publication). Second, abstracts that do not convert to a peer-reviewed publication can reflect weaker studies that authors feel are useful pilot data but not fit for publication or studies with methodological issues that are detected at peer review.

The wide variety of methodologies used in the papers included in this review precluded completion of a meta-analysis. After all, there are no agreed upon standards for performing health monitoring with either SBS or EHM. For example, SBS programs can vary in sentinel immune status (immunocompetent or compromised), numbers, timing, frequency of testing, type of testing (e.g., serology, PCR, histopathology, bacterial culture, parasitology), sample type, and ratio of colony cages to sentinel cages [69]. Similarly, EHM programs vary in methodology as indicated in Table 1 and in aspects related to frequency of testing, sample type, and ratio of colony cages to media.

Accordingly, it was necessary to turn more discrete outcome variables into yes/no detection, which could have elevated estimates of sensitivity. However, this would have affected SBS and EHM equally, still allowing a fair comparison. Furthermore, this also reflects the reality of health monitoring–a single positive result is typically interpreted as indicative of a broader pathogen presence. Despite these limitations, it is still clear that using EHM is superior to SBS for routine rodent health monitoring.

## Conclusion

Our findings highlight the strong evidence-base supporting the superiority of EHM over SBS regardless of pathogen type, pathogen importance level, or specific type of EHM. Of note, even as of 2023 there is a larger evidence base for both EDT and SFSB than there was for SBS in 2016 [1]. This clear superiority in testing efficacy, combined with the 3Rs benefit of Replacement of animals, argues for a scientific and ethical priority to replace SBS with EHM. Cost and labor savings only further the case for EHM.

## Supporting information

**S1 File. PRISMA checklist.** The 2020 PRISMA checklist with section references to where all relevant information can be found.
(DOCX)

**S2 File. A priori protocol.** The study protocol written out a priori before beginning data collection.
(PDF)

**S3 File. Search strategy.** The full text search strategy used for each database.
(DOCX)

**S1 Table. Pathogen evaluation and recode list.** A list of all pathogens evaluated in each article with conclusions as to whether they were detected and information on how they were coded for analysis.
(XLSX)

**S2 Table. Environmental health monitoring articles.** A detailed table of all articles included in the systematic literature review.
(XLSX)

**S3 Table. Study list.** A complete list of all articles screened in this systematic review, decision as to whether to include, and reason for decision.
(XLSX)

**S1 Dataset. Final dataset and analyses.** This file includes the SAS code for the dataset used in analysis and code for both final analyses as well as notes on the steps taken for each analysis.
(DOCX)

**S2 Dataset. Matched dataset and analyses.** This file includes the SAS code for the matched dataset only including the publications that directly compared EHM and SBS and the corresponding analyses and steps.
(DOCX)

## Acknowledgments

We would like to gratefully acknowledge Lauren Young for assistance with study selection and Jane Yacitilla for early consultation on our search strategy. We also thank the 3Rs Collaborative staff, volunteers, and members for making this research possible. We appreciate all the laboratory animal personnel who have been integral in helping their institutions replace soiled bedding sentinels with environmental health monitoring. Finally, we thank all of the sentinel rodents that have been, and are currently, used for rodent health monitoring programs. Their contribution has helped ensure the validity and reproducibility of scientific findings for several decades. We hope this review helps accelerate their Replacement where appropriate.

## Author Contributions

**Conceptualization:** Megan R. LaFollette, Kerith R. Luchins, Christopher A. Manuel, Patricia L. Foley, Wai H. Hanson, Christina Pettan-Brewer, Caroline B. Winn, Joseph P. Garner.

**Data curation:** Caroline S. Clement.

**Formal analysis:** Megan R. LaFollette, Joseph P. Garner.

**Investigation:** Megan R. LaFollette, Kerith R. Luchins, Christopher A. Manuel, Patricia L. Foley, Joseph P. Garner.

**Methodology:** Megan R. LaFollette, Caroline S. Clement, Joseph P. Garner.

**Project administration:** Megan R. LaFollette, Joseph P. Garner.

**Resources:** Megan R. LaFollette, Joseph P. Garner.

**Software:** Joseph P. Garner.

**Supervision:** Megan R. LaFollette, Joseph P. Garner.

**Validation:** Kerith R. Luchins, Christopher A. Manuel, Patricia L. Foley, Wai H. Hanson, Christina Pettan-Brewer, Caroline B. Winn.

**Visualization:** Megan R. LaFollette.

**Writing – original draft:** Megan R. LaFollette, Caroline S. Clement, Christopher A. Manuel, Caroline B. Winn, Joseph P. Garner.

**Writing – review & editing:** Megan R. LaFollette, Kerith R. Luchins, Christopher A. Manuel, Patricia L. Foley, Wai H. Hanson, Christina Pettan-Brewer, Caroline B. Winn, Joseph P. Garner.

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
