## [Decision Letter · Decision Letter 0]

7 Jun 2024

PONE-D-24-05570Do we still need a canary in the coal mine for laboratory animal facilities? A systematic review of environmental health monitoring versus soiled bedding sentinelsPLOS ONE

Dear Dr. LaFollette,

Thank you for submitting your manuscript to PLOS ONE. After careful consideration, we feel that it has merit but does not fully meet PLOS ONE’s publication criteria as it currently stands. Therefore, we invite you to submit a revised version of the manuscript that addresses the points raised during the review process.

We look forward to receiving your revised manuscript.

Kind regards,

I Anna S Olsson, Ph.D.

Academic Editor

PLOS ONE

Journal Requirements:

2. We are unable to open your Supporting Information file [S1 Dataset_Final Analysis, 20240117.sas and S2 Dataset_Matched Analysis, 20240117.sas]. Please kindly revise as necessary and re-upload.

Reviewers' comments:

Reviewer's Responses to Questions

**Comments to the Author**

1. Is the manuscript technically sound, and do the data support the conclusions?

Reviewer #1: Yes

2. Has the statistical analysis been performed appropriately and rigorously? 

Reviewer #1: Yes

3. Have the authors made all data underlying the findings in their manuscript fully available?

Reviewer #1: Yes

4. Is the manuscript presented in an intelligible fashion and written in standard English?

Reviewer #1: Yes

5. Review Comments to the Author

Reviewer #1: The work described in this manuscript is well received. It is about time this information gets analyzed properly and disseminated to the research community. Nonetheless a few edits are suggested. Please see attached document.

6. PLOS authors have the option to publish the peer review history of their article (what does this mean?). If published, this will include your full peer review and any attached files.

Reviewer #1: No

---

## [Author Response · Author response to Decision Letter 0]

18 Jul 2024

July 18, 2024

Dear Dr. Olsson,

Thank you for inviting a revision of the manuscript PONE-D-24-05570, “Do we still need a canary in the coal mine for laboratory animal facilities? A systematic review of environmental health monitoring versus soiled bedding sentinels.” We are grateful for the thoughtful feedback from the reviewers and believe their comments have strengthened the manuscript.

We have completed all revisions and addressed each comment individually below. The reviewers’ comments have been bolded and our responses are not bolded in the attached document.

We look forward to receiving your decision. Thanks for considering this manuscript revision.

Sincerely,

Megan LaFollette and co-authors.

Reviewer #1: 

The work described in this manuscript is well received. It is about time this information gets analyzed properly and disseminated to the research community. Nonetheless a few edits are suggested. Please see attached document for a formatted version of these comments.

We thank reviewer 1 for the overall positive comments. We look forward to dissemination and have considered the edits.

1. Great job and thank you for doing this work. 

Thank you.

2. Page 3 line 61: “Traditionally, rodent health monitoring programs have relied on sentinel rodents, that have an intact immune system and are not directly involved with experimental studies.” 

Some programs use immunodeficient mice in certain areas or situations or rely on so called “provocation tests” to immunosuppress SBS to activate a dormant infection. Suggest this topic be expanded here or the discussion section given you do mention the use of nude mice (line 393). 

Thank you for the reminder that immunodeficient mice are sometimes used as sentinels. We have modified the sentence on page 3, line 61 to only reference that sentinel rodents are not directly involved with experimental studies and then added a clarifying sentence to indicate that sentinels may or may not be used. “Typically, sentinel rodents have an intact immune system for pathogen detection by serology, but immunodeficient strains have also been used to better represent the population under surveillance or propagate the pathogen for easier detection by molecular methods.” 

We also elaborate further in the discussion section, lines 609-614 on the variation that exists in SBS methodology including choice of animals to be used. “This is seen through SBS programs which can vary in the choice of sentinel animals used (immunocompetent vs immunocompromised)...”

3. Page 3 line 66: “Sentinel animals are typically exposed at each cage-change, serially sampled, euthanized, and then replaced after 3-6 months [1]. Typically, there is one SBS cage per rack, or per side of a rack, both to give specificity as to the location of a pathogen and to increase sensitivity by avoiding dilution of the pathogen in too large of a pool of soiled bedding.” 

First, note there are descriptions of live sentinel programs using mice for up to 12 months. See Lab Anim (NY). 2003 May;32(5):36-43.doi: 10.1038/laban0503-36. Second while you describe a “typical” program, it would be good to mention there are no agreed upon standards for SBS programs and now neither for non-animal alternatives discussed here. I agree with the overall assessment, but it is important to point out to the readers every comparison must be evaluated considering very specific context. Perhaps the discussion section is best to touch on this topic. 

We agree with the reviewer that both SBS and EHM methods vary. We have expanded on this in the discussion section lines 611-616. “After all, there are no agreed upon standards for performing health monitoring with either SBS or EHM. For example, SBS programs can vary in sentinel immune status (immunocompetent or compromised), numbers, timing, frequency of testing, type of testing (e.g., serology, PCR, histopathology, bacterial culture, parasitology), sample type, and ratio of colony cages to sentinel cages. Similarly, EHM programs vary in method as indicated in Table 1, frequency of testing, sample type, and ratio of colony cages to media.”

4. Page 4 line 106: “As a related advantage, unlike SBS, EHM can reveal both active and past infections from which a colony has recovered [15].” 

I believe this statement to be incorrect. Overall, it really depends on testing frequency and the agent in question. SBS can detect active and past infections. For example, a parasitological assay such as anal tape test on a sentinel mouse can identify an active infection in the colony, and a serological assay can detect infections from which a colony may have recovered by the time a positive serological conversion is detected. 

We appreciate you catching this mistake, this sentence has been deleted. 

5. Page 4 line 107: “Second, because the sensitivity of EHM comes from the sensitivity and specificity of PCR based assays, EHM avoids the complexity of using differing methods of detection for different pathogens and can detect even minute levels of pathogen genomic sequences in samples collected from the environment [15].” 

Given every assay has a detection limit it is possible minute amounts of pathogen genomic sequences that could not be detected by EHM (PCR), could in fact be multiplied in an exposed SBS allowing detection, albeit somewhat delayed. Correct?

Although in principle this could potentially occur, we believe it is highly unlikely since if only minute amounts of sequences are present it would be highly unlikely for the dose to be infectious. The sentence from the manuscript is a summary of the Albers, et al. Manuscript that states: 

“By contrast, HM by conventional methodologies is typically reliant on the testing sentinels that are exposed to infectious agents in a colony through routine transfers of soiled bedding. This approach is problematic for several reasons. First, fomite transmission to sentinels is not effective for important host-adapted and environmentally labile pathogens.17,24,26,37,46 Moreover, soiled bedding might not transfer infection, even of an environmentally stable pathogen, if the dose to which sentinels are exposed is subinfectious because: 1) the prevalence of colony infection is low, as is common for cage-level barrier systems, or 2) sentinels are resistant to infection due to their age or genetic background.1,6,18,19,21,27 Finally, the sentinels may test positive after infections from sources other than the colony being monitored. For instance, adventitious infection of sentinels could have occurred prior to their placement while in transit or quarantine.50"

6. Page 4 line 110: “Third, EHM eliminates any animal welfare concerns as well as animal and labor costs inherent to a SBS program [22].” 

The statement makes it sound like there are no labor cost when using EHM, clearly not the case. 

Thank you for this catch. This has been rephrased to “Third, EHM eliminates any animal welfare concerns as well as removes animal costs and decreases labor costs inherent to a SBS program.”

7. Are there any benefits to SBS over EHM that can be discussed here? How about the ability to identifying adventitious agents for which there is no PCR assay, or it is not included in the test panel? What happens if a wild mouse carrying unknown agents enters a vivarium and exposes and infects colony animals with agents not included in the EHM PCR panel? Biosecurity failures aside, how would one find such agents? SBS might not find them either but there are examples in the literature that show this has happened. 

Thank you for this important point. In these cases, we believe that a thorough diagnostic work-up would be necessary on colony animals. We have addressed this in the Discussion section starting on line 527: “Although we recommend EHM as the core part of rodent health monitoring, it should not be considered a substitution for techniques required to detect and characterize novel agents. After all, as EHM is reliant on PCR it can miss any agents that are novel or not included in the testing panel. Therefore, if abnormal clinical signs or study results are observed in colony rodents without a corresponding diagnosis via the EHM panel, we recommend performing a thorough diagnostic workup (e.g., histopathology, clinical chemistry, blood count) on the affected colony animals.”

8. Page 12, line 274: The acronym GEE appears twice in the manuscript, but it is not defined. Please do so. 

The acronym has been replaced with General Estimating Equations in both instances. 

9. I agree with the overall conclusions, but the manuscript seems a bit biased towards non-animal alternatives to SBS. It avoids any mention of positive attributes of SBS programs some of which I have mentioned before and perhaps should be included in the discussion section. I suggest you temper the tone to avoid alienation of the more traditional minded readers.

One item I thought was lacking in the evaluation of the literature comparing SBS with non-animal alternatives is the fact the assays used on SBS on those papers were not indicated in the supplemental materials. For example, the S1_Table_Pathogens does not specify which methods were used to detect each agent listed when using SBS. While non-animal alternatives rely solely on PCR, SBS can be tested using serological methods, PCR (skin, oral, feces, organs), bacterial cultures, traditional parasitological assays (scrape, floats, wet mounts), and histological evaluations. All these impacts, to a great degree, the detection rates and therefore the overall scoring to SBS programs. It has been shown, for example, that PCR on SBS sentinels is better than fecal float to detect some pinworms, and a combination of fecal PCR plus cecal examination is better than fecal PCR alone. 

This is something I suggest should be discussed, perhaps as another factor that complicates these types of comparisons.

We thank the reviewer for their perspective on this topic. However, we do not feel we are inaccurately presenting the data as biased towards non-animal alternatives; rather we are presenting the results fairly and accurately. Indeed, that is the whole point of the formal systematic review process – to summarize the literature fairly, transparently, and without bias. Based upon the results of this review, we see no overall positive attributes to SBS programs when considering the superiority of EHM methods in their pathogen detection, benefits to animal welfare, and replacement of animals in accordance with the 3Rs.

Nevertheless, accommodating earlier comments has resulted in softening of the language in many places, which hopefully addresses the general point of avoiding the appearance of bias.

Furthermore, although we agree that using PCR methods with SBS is superior to other SBS techniques, comparing different types of SBS was not the goal of this review and is considered out of scope. Rather, our goal was to directly compare SBS to various distinct EHM methods. Many SBS programs use a combination of methods making it difficult to characterize them into distinct methods. Therefore, we have declined to expand S1_Table.

We believe this comment has been addressed in the limitations section by acknowledging there are no agreed upon standards and programs greatly vary. For this reason, it was necessary to dichotomize the data into yes/no detection. However, this simplification equally affects the results of SBS and EHM therefore allowing a fair comparison and reflecting the reality that a single positive is interpreted as a broader indication of pathogen presence. Finally, we have added an additional figure to clearly explain how much more effective EHM is compared to SBS. This figure maps only head-to-head comparisons of EHM and SBS and lists the pathogens detected by each where the other failed. It shows that when EHM detected a pathogen, SBS failed in 29% of cases (compared to EHM failing in only 3% of cases where SBS detected a pathogen).

---

## [Decision Letter · Decision Letter 1]

9 Sep 2024

PONE-D-24-05570R1Do we still need a canary in the coal mine for laboratory animal facilities? A systematic review of environmental health monitoring versus soiled bedding sentinelsPLOS ONE

Dear Dr. LaFollette,

Thank you for submitting your manuscript to PLOS ONE. After careful consideration, we feel that it has merit but does not fully meet PLOS ONE’s publication criteria as it currently stands. Therefore, we invite you to submit a revised version of the manuscript that addresses the points raised during the review process.

Please see the request to correct a typo Page 14 line 336 should say FELASA, not FELESA.

We look forward to receiving your revised manuscript.

Kind regards,

I Anna S Olsson, Ph.D.

Academic Editor

PLOS ONE

Journal Requirements:

Reviewers' comments:

Reviewer's Responses to Questions

**Comments to the Author**

1. If the authors have adequately addressed your comments raised in a previous round of review and you feel that this manuscript is now acceptable for publication, you may indicate that here to bypass the “Comments to the Author” section, enter your conflict of interest statement in the “Confidential to Editor” section, and submit your "Accept" recommendation.

Reviewer #1: All comments have been addressed

2. Is the manuscript technically sound, and do the data support the conclusions?

Reviewer #1: Yes

3. Has the statistical analysis been performed appropriately and rigorously? 

Reviewer #1: Yes

4. Have the authors made all data underlying the findings in their manuscript fully available?

Reviewer #1: Yes

5. Is the manuscript presented in an intelligible fashion and written in standard English?

Reviewer #1: Yes

6. Review Comments to the Author

Reviewer #1: I commend the authors for the work culminating with this manuscript which I know will be well received by the targeted community.

One tiny correction is required: Page 14 line 336 should say FELASA, not FELESA.

7. PLOS authors have the option to publish the peer review history of their article (what does this mean?). If published, this will include your full peer review and any attached files.

Reviewer #1: No

---

## [Author Response · Author response to Decision Letter 1]

12 Sep 2024

We thank the reviewer for their positive feedback and attention to detail. We have corrected the typo and attached a corrected manuscript.

---

## [Editor Report · Decision Letter 2]

25 Sep 2024

Do we still need a canary in the coal mine for laboratory animal facilities? A systematic review of environmental health monitoring versus soiled bedding sentinels

PONE-D-24-05570R2

Dear Dr. LaFollette,

We’re pleased to inform you that your manuscript has been judged scientifically suitable for publication and will be formally accepted for publication once it meets all outstanding technical requirements.

Kind regards,

I Anna S Olsson, Ph.D.

Academic Editor

PLOS ONE
---

## [Editor Report · Acceptance letter]

25 Nov 2024

PONE-D-24-05570R2 

PLOS ONE

Dear Dr. LaFollette, 

I'm pleased to inform you that your manuscript has been deemed suitable for publication in PLOS ONE. Congratulations! Your manuscript is now being handed over to our production team.

Kind regards, 

on behalf of

Dr. I Anna S Olsson 

Academic Editor

PLOS ONE